# Soil fertility mapping of a cultivated area in Resunga Municipality, Gulmi, Nepal

**Prabin Ghimire**[1], **Santosh Shrestha**[1,2], **Ashok Acharya**[3], **Aayushma Wagle**[4], **Tri Dev Acharya**[5]*

**1** Department of Soil Science, Institute of Agriculture and Animal Science, Tribhuvan University, Kirtipur, Bagmati, Nepal, **2** Agricultural Technology Center, Lalitpur, Bagmati, Nepal, **3** Department of Soil Science and Agri-Engineering, Agriculture and Forestry University, Rampur, Bagmati, Nepal, **4** Department of Agronomy, Agriculture and Forestry University, Chitwan, Bagmati, Nepal, **5** Institute of Transportation Studies, University of California, Davis, Davis, California, United States of America

* tdacharya@ucdavis.edu

**Data Availability Statement:** With the permission of Resunga Municipality, Nepal soil sampling data can be made available by the first author or information officer of Agricultural Knowledge Center Gulmi, Tamghas, Nepal (https://gulmi.akc.

## Abstract

Soil fertility maps are crucial for sustainable soil and land use management system for predicting soil health status. However, many regions of Nepal lack updated or reliable soil fertility maps. This study aimed to develop the soil fertility map of agricultural areas in Resunga Municipality, Gulmi district of Nepal using the geographical information system (GIS) technique. A total of 57 composite geo-referenced soil samples from the depth (0–20 cm) were taken from the agricultural land of an area of 52 km$^2$. Soil samples were analyzed for their texture, pH, organic matter, total nitrogen, available phosphorous, available potassium, available boron, and available zinc. These parameters were modelled to develop a soil quality index (SQI). Using the kriging tool, obtained parameters were interpolated and digital maps were produced along with soil quality and nutrient indices. The result showed that the study area lies within the fair (0.4 to 0.6) and good (0.6 to 0.8) range of SQI representing 96% and 3% respectively. Soil organic matter and nitrogen showed moderate variability exhibiting a low status in 95% and 86% of the total study area. Phosphorous and potassium showed medium status in 88% and 75% of the study area, respectively. Zinc was low and boron status was medium in most of the area. To maintain soil fertility is by improving the rate of exogenous application of fertilizers and manures. The application of micronutrients like boron and zinc is highly recommended in the study area along with organic manures. The soil fertility map can be used as a baseline for soil and land use management in Resunga Municipality. We recommend further studies to validate the map and assess the factors affecting soil fertility in this region. Soil fertility maps provide researchers, farmers, students, and land use planners with easier decision-making tools for sustainable crop production systems and land use management systems.

## Introduction

Land and soil have directly or indirectly been included in the global policy framework of the United Nations (UN) Sustainable Development Goals (SDGs). Sustainable soil management is

gov.np) via phone at +977-079-520126 or email at akcgulmi2075@gmail.com. Other results and maps generated during this study are included in the article.

**Funding:** The authors received no specific funding for this work. The APC was funded by the University of California Davis Open Access Fund (UCD-OAF).

**Competing interests:** The authors have declared that no competing interests exist.

an important aspect of achieving the SDGs, among which ten targets have a direct relation to soil. Soil quality improvement is requisite for attaining the SDGs, specifically the goals of zero hunger (SDG 2), climate action (SDG 13), and life on land (SDG 15) [1]. The unavailability of soil-based indicators assigned to soil-related SDGs is a problem. So, there is a need for soil-based indicators from signalling, design, implementation, and evaluation to achieve the SDGs [2]. Soil fertility and quality refer to the nutrient-supplying ability of soil which is the important aspect that defines the soil and agricultural productivity [3]. Furthermore, land degradation (SDG 15.3) is the consequence of the loss of soil productivity [4]. The set of measurable attributes termed indicators including the nutrient status and environmental condition can be used for monitoring and evaluating soil fertility [5].

The measurable indicators like nutrient status, organic carbon are suggested to achieve the soil-related SDGs target to implement resilient agricultural practices for increasing production and productivity and maintaining the ecosystem, and strengthening the capacity for climate change adaptation which progressively improves the land and soil by 2030, ensuring the sustainable food production system [1].

Soil fertility and quality determination and assessments are essential for the optimization and sustainability of the agroecosystem [6]. It plays an important role in land use planning, resource management, and site analysis [7]. Soil data that includes physical and chemical properties are needed to access and monitor which impacts the soil fertility and quality with an emphasis on nutrient status and organic carbon to achieve the SDGs [8]. Digital soil mapping (DSM) provides detailed work and can be managed and stored within a geographic information system (GIS). The traditional soil survey and mapping are based on the recognition of soil properties and their quantitative relationship within the landscape and environmental variables on conceptual models [9]. The implementation of sustainable land use management requires soil mapping [10] and digital soil mapping (DSM) uses an integration of soil data and its environmental covariates. It requires field sampling, laboratory analysis, and remote sensing arranging in various spatial and temporal scales by adopting the appropriate support system [9].

There is a need the provide specific and detailed soil information and use suitable geostatistical methods to study the spatial structures. The use of techniques of spatial interpolation known as kriging to estimate the variable at the unsampled location. Kriging provides an unbiased and linear estimate of variables, and the most common use of kriging is ordinary kriging, it testifies that geostatistical interpolation techniques can be used for the regional level even without ancillary data if there is sufficient data within the study area [11].

Nepal is an agricultural country and soil resources are considered an integral part of its economy. Variability in soil fertility and quality is found across Nepal mainly due to its forming factors such as parent material, climate, topography, and soil organisms [12]. Despite the advancement of digital soil mapping around the world, there are narrative reports and soil maps in the analogue format generated from the majority of soil survey work in Nepal. Soil and Terrain (SOTER) database by Food and Agriculture Organization (FAO) in collaboration with the Nepal survey department has developed a database at the scale of 1: million. This database lacks the details to be used at local levels by farmers. The gap of high-resolution maps with spatial distribution in soil physical and chemical properties can be filled by field sampling and providing descriptive statistics and spatial variability of soil fertility parameters, soil quality and soil nutrient index in the form of maps that will have actual value to the farmers' fields [13]. Nepal lacks these types of studies with detailed spatial variability, as the hilly regions have high variability from place to place [14].

The objective of the study is to perform soil sampling and provide descriptive statistics and spatial variability of soil fertility parameters, soil quality and soil nutrient index in the form of

maps of a cultivated area in Resunga Municipality, Gulmi, Nepal. The study uses a simple and widely used methodology of ordinary kriging for interpolation of soil nutrient variability that can be used to develop soil maps. It is the first attempt at high-resolution mapping in the region, the maps produced will immensely benefit the farmers and local authorities for sustainable soil and land use management and will eventually aid in obtaining the SDGs in the region.

## Materials and methods

### Study area

The study area, Fig 1, was the Resunga municipality of Gulmi district, Nepal. It lies between 637–2338 mean average sea level with a total 85 km$^2$ area and approximately 52 km$^2$ and 32 km$^2$ of agricultural and forested area respectively. It experiences an annual average rainfall of 1395 mm and a temperature range of 14–40°C. The study area had citrus cultivation, maize-seasonal vegetables in upland areas, and rice-wheat-maize in lowlands. This region is affected by rapid urbanization, agricultural land is being gradually reduced and transformed into construction sites. The preparation of a soil fertility map shows the present status of agricultural land and can provide the key information for its protection and the reclamation for sustainable land management to ensure sustainable crop production.

### Soil sampling

Given the importance of soil fertility parameters for agricultural production in the study area, sample points were collected from the agricultural area only excluding the forested area. The study area was classified based on slope, aspect, and elevation of the area, and random sampling was done proportionately to represent the complete geomorphology of the municipality. Being a hilly region, sampling was challenging in the area with a high slope. The randomly sampled points at high slopes were mostly rocky surfaces, so the samples were taken from nearby land with soil.

The work was done in technical support of the Agriculture Knowledge Center, Gulmi and Soil and Fertilizer Testing Laboratory, Kanchanpur. Being both a government organization and part of their ongoing soil mapping work, this work required no permits for soil sampling and testing. A total of 57 samples were taken from the agricultural area only. A Garmin GPS device was used to reach the points and samples were taken from 0–20 cm depth with the help of a soil auger a profile study was also performed. Some glimpses of the work are shown in S1 Fig in S1 File.

Collected soil samples were prepared and analyzed at the regional soil and fertilizer testing laboratory, Kanchanpur, Mahendranagar, Nepal for routine parameters i.e., soil pH, soil texture, total nitrogen, available phosphorus and potassium, soil organic matter (SOM), and micronutrients such as boron and zinc following methods given in Table 1.

### Methodology

Soil chemical properties data of each parameter (pH, OM, N, $P_2O_5$, $K_2O$, SQI, B, and Zn) were analyzed for descriptive statistics (mean, median, maximum, minimum, standard deviation, standard error of the mean, and coefficient of variation) using R studio 4.0.1. A fertility parameter is considered as weakly variable, moderately variable, or strongly variable when the coefficient of variation (CV%) is < 10, 10–100 and > 100, respectively [22].

Soil fertility and soil quality map were prepared with ArcGIS 10.8 using interpolation techniques. The ordinary kriging method was deployed in interpolation from the geostatistical

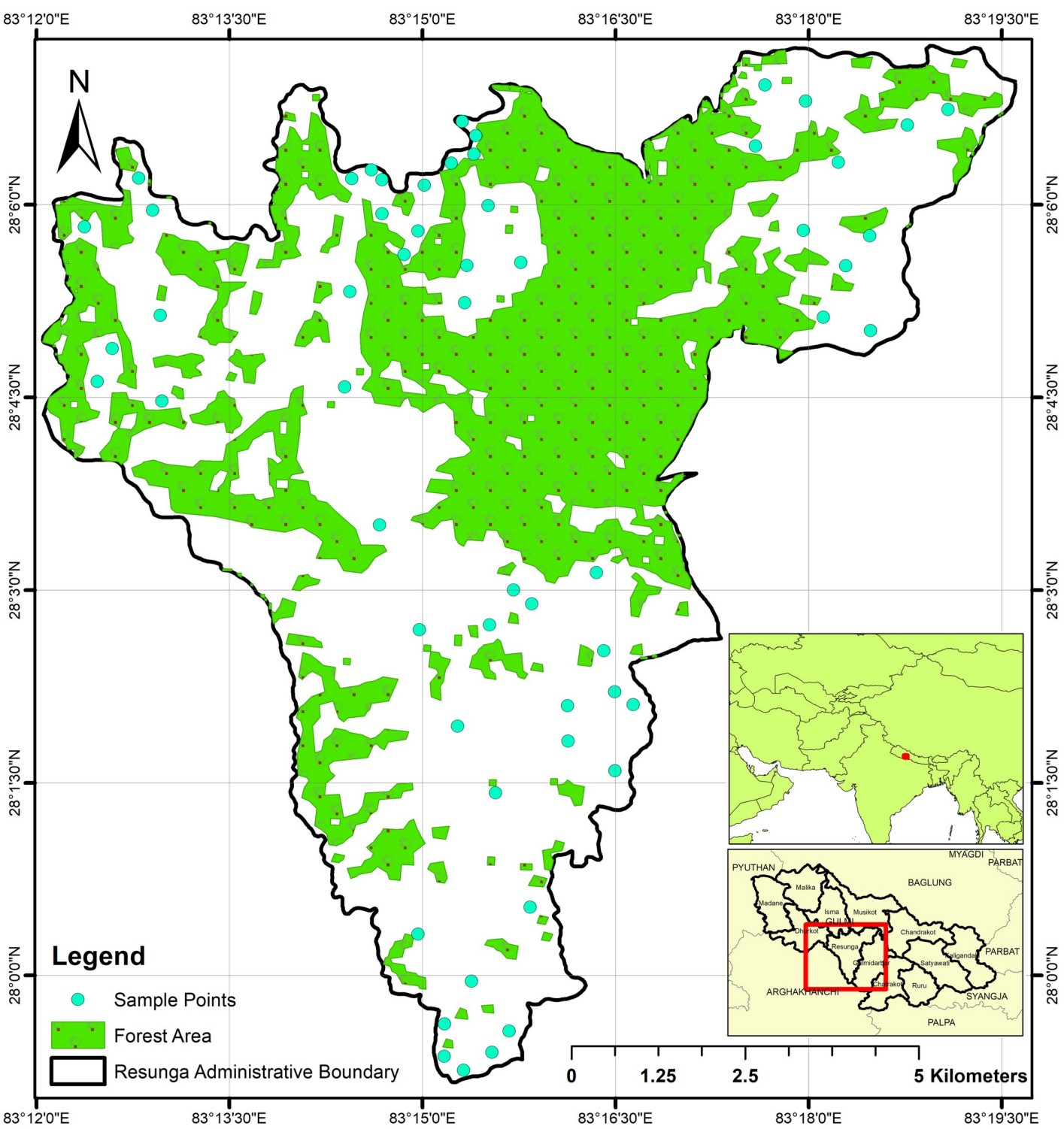

**Fig 1. Location map of Resunga municipality of Gulmi district, Nepal as a study area.**

**Table 1. Soil parameters were tested, and methodology was adopted.**

| S. N. | Parameter | Unit | Methods |
|---|---|---|---|
| 1 | Total nitrogen (N) | % | Kjeldahl method [15] |
| 2 | Available phosphorus ($P_2O_5$) | ppm | Modified Olsen's Bicarbonate Method [16] |
| 3 | Available potassium ($K_2O$) | ppm | Neutral Ammonium Acetate method [17] |
| 4 | Soil pH | - | 1:2.5 Potentiometric method [17] |
| 5 | Soil organic matter (SOM) | % | Walkley Black method [18] |
| 6 | Soil texture | - | Hydrometer method [19] |
| 7 | Available boron (B) | ppm | Hot water extraction method [20] |
| 8 | Available zinc (Zn) | ppm | DTPA extraction method [21] |

Note: ppm is parts per million and DTPA is diethylenetriaminepentaacetic acid.

wizard for nitrogen, phosphorus, potassium, zinc, boron, pH, and soil quality index (SQI). Thiessen polygon was deployed for soil texture map preparation. Fig 2 shows the overall workflow adopted in the study.

**Soil quality index (SQI).** The soil parameter like available water capacity, infiltration capacity, and aggregate stability are included in the development of the SQI [23, 24] but are not easily available in many regions due to the lack of resources and equipment. So, in Nepal, the soil quality rating (SQR) system was developed by Bajracharya *et al* [25] for the four major soil parameters by adding the product of the weighting factor with assigned parameter ranking values from 0 to 1 (0 is the worst and 1 is best). This is a simple and readily applicable semi-quantitative approach for assessing the overall relative soil quality from an agricultural perspective. Nepal Agriculture Research Council (NARC) and the Department of Agriculture, Nepal developed the weighted value for each parameter considering various types of soils in the country [25]. According to NARC, soil organic matter is considered an important factor as it influences nutrient availability; aggregate stability, erosion susceptibility, etc. so it was given the value of 0.4 (c), the major nutrients like N, P, and K are important for crop production so it is valued as 0.3 (d), soil texture as 0.2 (a) and pH is considered the as low degree of importance, but essential soil quality so valued as 0.1(b). Thus, the final SQI [26] was calculated with the formula given below:

$$SQI = \left[ (a^*R_{STC}) + \left( b^*R_{pH} \right) + (c^*R_{oc}) + (d^*R_{NPK}) \right] \tag{1}$$

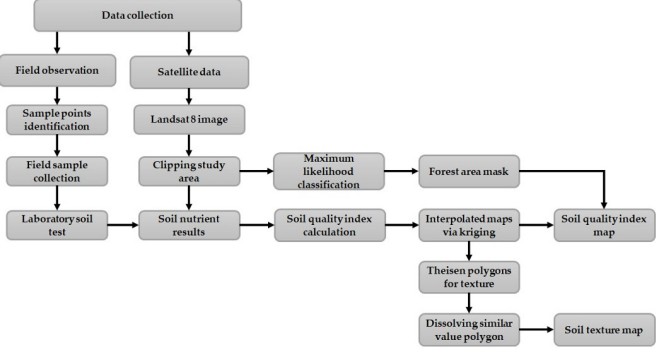

**Fig 2. Workflow of the methodology adopted.**

where, a, b, c, and d are the weighted values of each parameter which are 0.2, 0.1, 0.4, and 0.3, respectively.

$R_{STC}$, $R_{pH}$, $R_{oc}$, and $R_{NPK}$ are assigned ranking values of texture, pH, organic matter, and NPK respectively which are given in Table 2. The scoring method was also developed by the NARC and was used to interpret the SQI.

**Soil nutrient index (SNI).** The soil nutrient index (SNI) [27, 28] was calculated with the given formula:

$$SNI = \{(1^*A) + (2^*B) + (3^*C)\}/TNS, \tag{2}$$

where A is the number of samples in low categories, B is the number of samples in medium categories, C is the number of samples in high categories as per the classification of parameters in (Table 3), and TNS = Total number of samples. For any given soil parameter, SNI is low, medium, and high category if the index value is less than 1.67, the fertility status is low and the value between 1.67–2.33 and the status is medium. If the value is greater than 2.33, the fertility status is high.

**Spatial autocorrelation.** The global Moran I's index was used to calculate the spatial auto-correlation. The index is based on feature location and feature values. It evaluates whether the pattern is clustered, dispersed, or random. It utilizes the z-score and p-value to evaluate the significance of the index [29].

## Results

### Descriptive statistics of soil properties

The descriptive statistical summary for pH, organic matter, nitrogen, phosphorous, SQI, boron, and zinc are presented in Table 4. The coefficient of variation (CV) is used to interpret the variability. The CV ranges from 6.30% in pH to 100.09% in $P_2O_5$. There is a different degree of heterogeneity among the soil properties in the study area.

### Spatial variability map of soil properties

**a. Total nitrogen**: The concentration of N varied from 0.006% to 0.19% having a mean value of 0.08±005% (Table 4). N status was low (0.05–0.01%) in 95% of the study area (Table 5) (Fig 3a). The observed variability for total nitrogen content was moderate (40.83%) (Table 4).

**b. Available phosphorous**: The concentration of available $P_2O_5$ ranged from 0.66 to 120.72 ppm with a mean value of 20.96±2.78 ppm% (Table 4). Medium status (13–49 ppm)

**Table 2. Soil quality index (SQI) ranking values.**

| S. N. | Parameter | Ranking Value | | | | |
|---|---|---|---|---|---|---|
| | | 0.2 | 0.4 | 0.6 | 0.8 | 1.0 |
| 1 | Soil fertility (NPK) | Low | Mod. Low | Moderate | Mod. High | High |
| 2 | Soil pH | <4, >8.5 | 4–5 | 5–6 | 6–6.5, 7.5–8.5 | 6.5–7.5 |
| 3 | Soil organic matter | ≤0.5 | 0.5–1 | 1–2 | 2–4 | >4 |
| 4 | Soil textural class | C, S | CL, SC, SiC | Si, LS | L, SiL, SL | SiCL, SCL |
| 5 | Soil quality index | V. poor | Poor | Fair | Good | Best |

Note: C- Clay, S-Sand, CL-Clay loam, SC- Sandy Clay, SiC- Silty Clay, Si-Silt, LS-Loamy sand, SiL- Silty loam, SL-Sandy loam, LS Loamy Sand, L- loam, SL-Sandy loam, SiCL-Silty clay loam, SCL- Sandy Clay loam.

**Table 3. Soil nutrient index (SNI) ranges and remarks.**

| Soil nutrient index (SNI) ranges | Remarks |
| --- | --- |
| Below 1.67 | Low |
| 1.67–2.33 | Medium |
| Above 2.33 | High |

of available phosphorus in 88.02% of the area followed by high (25–49 ppm) in 11.38% of the study area was observed (Fig 3b) (Table 5). Available $P_2O_5$ showed strong variability (100.09%) among the studied soil samples.

**c. Available potassium**: The concentration of available $K_2O$ ranged from 13.61 to 346.34 ppm with a mean of 101.38±9.71 ppm (Table 4). This indicates a medium (49–125 ppm) to high (125–223 ppm) status of available potassium in most of the area (Table 5) (Fig 3c). The studied soil samples were moderate in variability (72.33%) (Table 4).

**d. Soil pH**: The soil pH varied from 5.30 to 7.50 having a mean of 6.80±0.06 (Table 4). The observed soil pH of the study area is slightly acidic representing 7% to neutral 91% of the total study area (Table 5) (Fig 3d). The soil pH showed weak variability (6.30%) among the studied samples.

**e. Soil organic matter**: The concentration of organic matter in all samples ranged from 0.09 to 2.48% with a mean value of 1.23±0.06% (Table 4). This indicates very low (<1%) in 13% of the study area to low (1–2.5%) status organic matter in 86% of the study areas (Table 5 (Fig 4a). The studied samples revealed the organic matter content in the soil has moderate variability of 34.13%.

**f. Soil texture**: The sand content in the studied samples ranged from 4.8 to 46.7% with a mean of 26.26±1.51%. Silt content was 25.8 to 50.3% with a mean of 36.3±0.82% while clay content ranged from 20.8 to 52% with a mean of 37.45±1.06% (Table 4). The five textural classes; loam, clay loam, silty clay loam, silty clay, and clay were determined, and most of the area contained clay loam (Fig 4b). The moderate variability of sand, silt, and clay was studied at 43.65%, 17.06%, and 21.34% respectively (Table 4).

**g. Available boron**: The concentration of boron ranged from 0.04 to 1.41 ppm with a mean of 0.44±0.04 (Table 4). This exhibits low (0.2–0.5 ppm) to medium (0.5-1ppm) content of available boron in 43% and 55% of the study area respectively (Table 5) (Fig 4c). Moderate variability of 74.75% was seen in the available boron concentration in sampled soil.

**Table 4. Descriptive statistics soil physiochemical status of Resunga Municipality, Gulmi, Nepal.**

| Descriptive Statistics | Soil Physiochemical Parameters | | | | | | | | | | |
| --- | --- | --- | --- | --- | --- | --- | --- | --- | --- | --- | --- |
| | N | $P_2O_5$ | $K_2O$ | pH | SOM | B | Zn | SQI | Sand | Silt | Clay |
| | % | ppm | ppm | | % | ppm | ppm | | % | % | % |
| Minimum | 0.006 | 0.66 | 13.61 | 5.30 | 0.09 | 0.04 | 0.01 | 0.28 | 4.8 | 25.8 | 20.8 |
| Maximum | 0.19 | 120.72 | 346.34 | 7.50 | 2.48 | 1.41 | 0.42 | 0.76 | 46.7 | 50.3 | 52.6 |
| Mean | 0.08 | 20.96 | 101.38 | 6.80 | 1.23 | 0.44 | 0.11 | 0.52 | 26.26 | 36.3 | 37.45 |
| Median | 0.09 | 15.344 | 79.56 | 6.90 | 1.29 | 0.41 | 0.09 | 0.54 | 24.8 | 33.6 | 38.2 |
| Standard Deviation (SD) | 0.03 | 20.98 | 73.33 | 0.43 | 0.4 | 0.33 | 0.07 | 0.1 | 11.46 | 6.19 | 7.99 |
| Standard Error of Mean (SEM) | 0.005 | 2.78 | 9.71 | 0.06 | 0.06 | 0.04 | 0.009 | 0.01 | 1.51 | 0.82 | 1.06 |
| CV % | 40.83 | 100.09 | 72.33 | 6.30 | 34.13 | 74.57 | 67.31 | 19.97 | 43.65 | 17.06 | 21.34 |

**Table 5. Areas of soil categories based on soil physio-chemical properties.**

| Parameter | Unit | Range | Class | Area (ha) | % of the Total Area |
|---|---|---|---|---|---|
| N | % | <0.05 | Very low | - | - |
| | | 0.05–0.10 | Low | 81 | 95% |
| | | 0.10–0.20 | Medium | 3.67 | 4% |
| | | 0.20–0.40 | High | - | - |
| | | >0.40 | Very high | - | - |
| $P_2O_5$ | ppm | <4 | Very low | - | - |
| | | 4–13.0 | Low | 0.07 | - |
| | | 13–49 | Medium | 74.91 | 88.02% |
| | | 25–49 | High | 9.69 | 11.38% |
| | | >49 | Very high | - | - |
| $K_2O$ | ppm | <25 | Very low | 0.09 | 0 |
| | | 25–49 | Low | 2.39 | 3% |
| | | 49–125 | Medium | 63.56 | 75% |
| | | 125–223 | High | 16.61 | 20% |
| | | >223 | Very high | 2.01 | 2% |
| pH | | <4.5 | Strongly acidic | 0.10 | - |
| | | 4.5–5.5 | Moderately acidic | 0.90 | 1% |
| | | 5.5–6.5 | Slightly acidic | 6.00 | 7% |
| | | 6.5–7.5 | Neutral | 77.67 | 91% |
| | | >7.5 | Strongly alkaline | - | - |
| SOM | % | <1 | Very low | 11.12 | 13% |
| | | 1–2.5 | Low | 73.56 | 86% |
| | | 2.5–5.0 | Medium | - | - |
| | | 5.0–10.0 | High | - | - |
| | | >10.0 | Very high | - | - |
| Soil texture | | | Loam | 11.21 | 13% |
| | | | Clay loam | 35.99 | 42% |
| | | | Silty clay loam | 6.68 | 8% |
| | | | Silty clay | 16.88 | 20% |
| | | | Clay | 14.34 | 17% |
| B | ppm | <0.2 | Very low | 1.74 | 2% |
| | | 0.2–0.5 | Low | 36.22 | 43% |
| | | 0.5–1 | Medium | 46.70 | 55% |
| | | 1–2 | High | - | - |
| | | >2.0 | Very high | - | - |
| Zn | ppm | <0.25 | Very low | 80.06 | 94.07% |
| | | 0.25–0.5 | Low | - | - |
| | | 0.5–1.0 | Medium | - | - |
| | | 1.0–2.0 | High | - | - |
| | | >2.0 | Very high | - | - |
| Soil Quality | | <0.2 | Very poor | - | - |
| | | 0.2–0.4 | Poor | - | - |
| | | 0.4–0.6 | Fair | 82.01 | 96% |
| | | 0.6–0.8 | Good | 2.66 | 3% |
| | | 0.8–1 | Best | - | - |

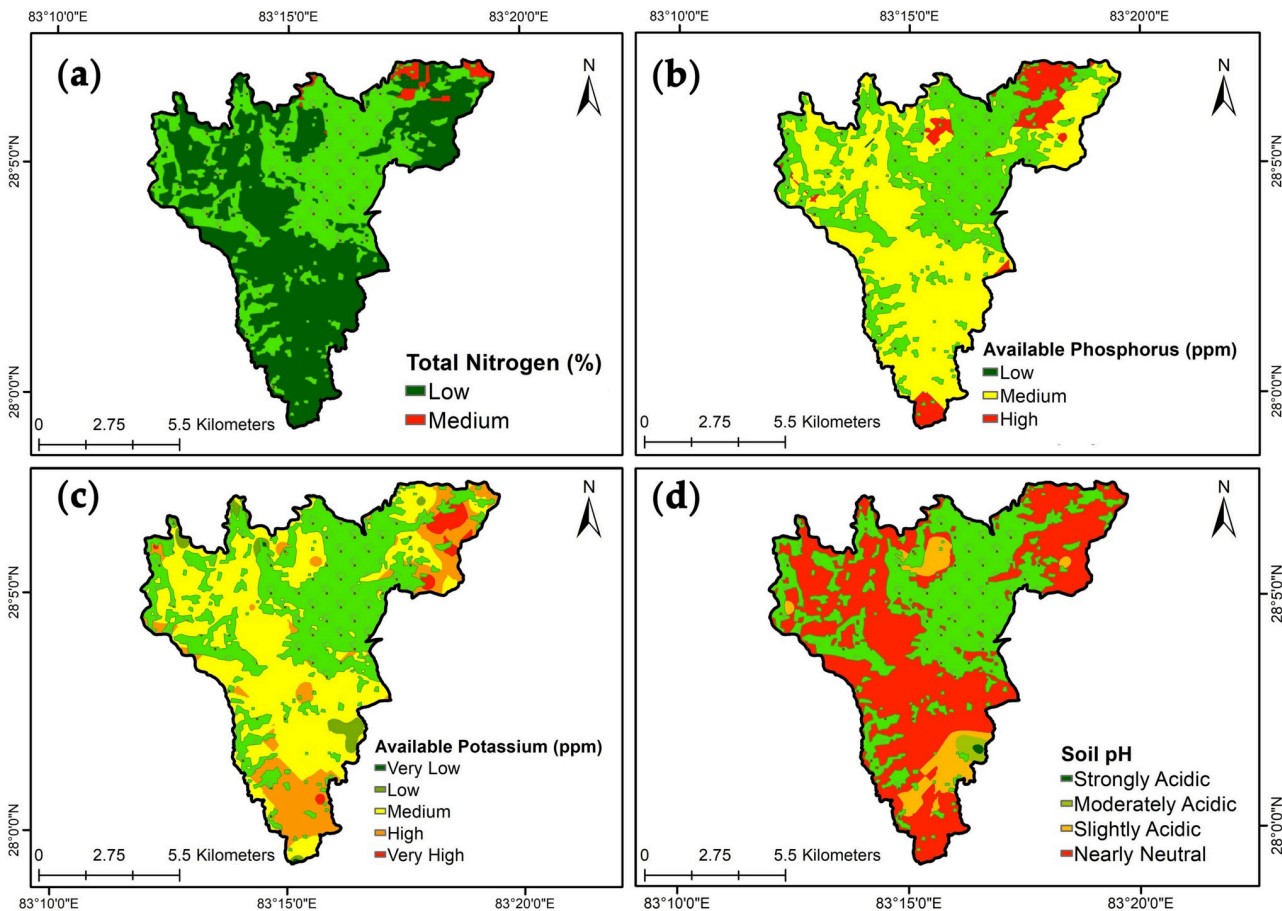

**Fig 3. Spatial variability and map of soil properties: (a) Total Nitrogen, (b) Available phosphorous, (c) Available potassium, and (d) soil pH.**

**h. Available zinc**: The concentration of available zinc ranged from 0.01 to 0.42 ppm with a mean of 0.11±0.009 ppm (Table 4). The observation was a very low (<0.25 ppm) status of available zinc throughout the study area (Table 5) (Fig 4d). The variability status was moderate (67.31%) in the sampled soil.

## Soil quality index (SQI)

The analysis of the soil quality index (SQI)in all samples exhibited in the range of 0.28 to 0.76 with a mean of 0.52±0.01 (Table 4). This shows fair (0.4–0.6) to good (0.6–0.8) soil quality status in 96% and 3% of the study area respectively (Table 5) (Fig 5). The SQI showed moderate variability (19.97%) in the studied soil samples.

## Soil nutrient index (SNI)

The SNI of N, P, K, and B are 1.368, 1.842, 2.053, and 1.439 respectively (Table 6). However, the SNI for soil organic matter and zinc was least i.e., 1. After calculating the SNI and comparing it to the SNI rating, the result shows that the soil samples were low in total nitrogen, soil organic matter, boron, and zinc; medium in available phosphorus and available potassium.

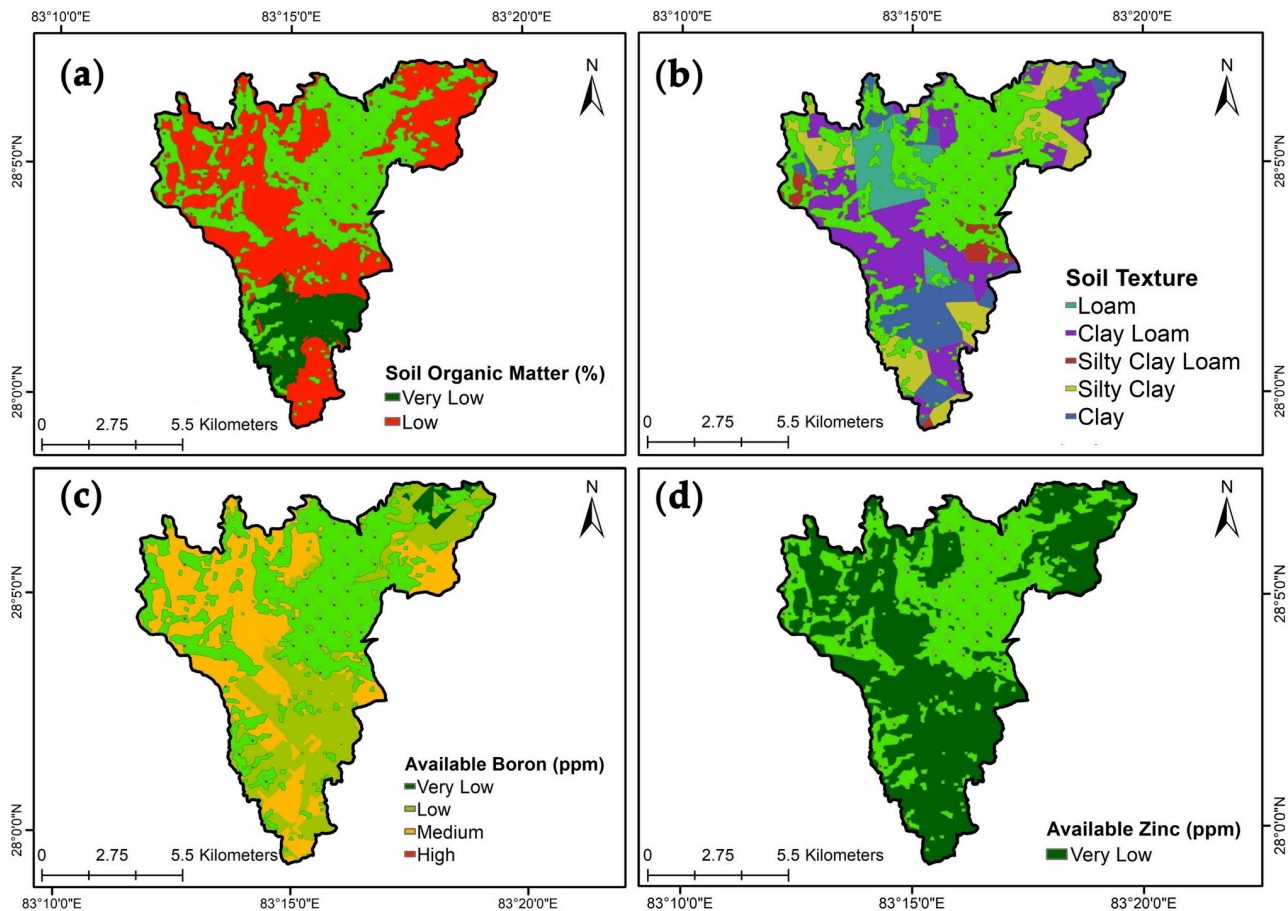

**Fig 4. Spatial variability and map of soil properties: (a) Soil organic matter, (b) Soil texture, (c) Available boron, and (d) Available zinc.**

## Spatial autocorrelation

Global Moran's I Index is used for the calculation of the spatial pattern of soil chemical properties (Table 7). The distribution of soil chemical properties was random throughout the study area is the hypothesis for the global Moran's I index analysis. With the small $p$-value ($p<0.005$), and the z score of very high or very low ($1.96<z$ and $z<-1.96$), a random form of distribution is not observed in the spatial pattern. the neighboring values are similar and have a spatial dependency when Moran's I Index value is positive. A negative Moran's I Index value suggests that the neighboring values are dissimilar and have a negative spatial dependency. Moran's I Index value of 0 suggests a lack of spatial pattern [30, 31]. Organic matter and $P_2O_5$ have a negative Moran's I Index, while all other soil chemical properties have a positive Moran's I Index for their spatial distribution. The spatial distribution of pH, OM, N, $P_2O_5$, $K_2O$, Zn, and B did not observe a significantly different from a random distribution at $p<0.005$.

## Discussion and conclusion

This study focused on developing soil fertility maps in the Resunga Municipality of Gulmi district of Nepal. Based on the maps we obtained, there is spatial variability in soil fertility parameters across the study site. The ordinary kriging tool was used to interpolate those parameters

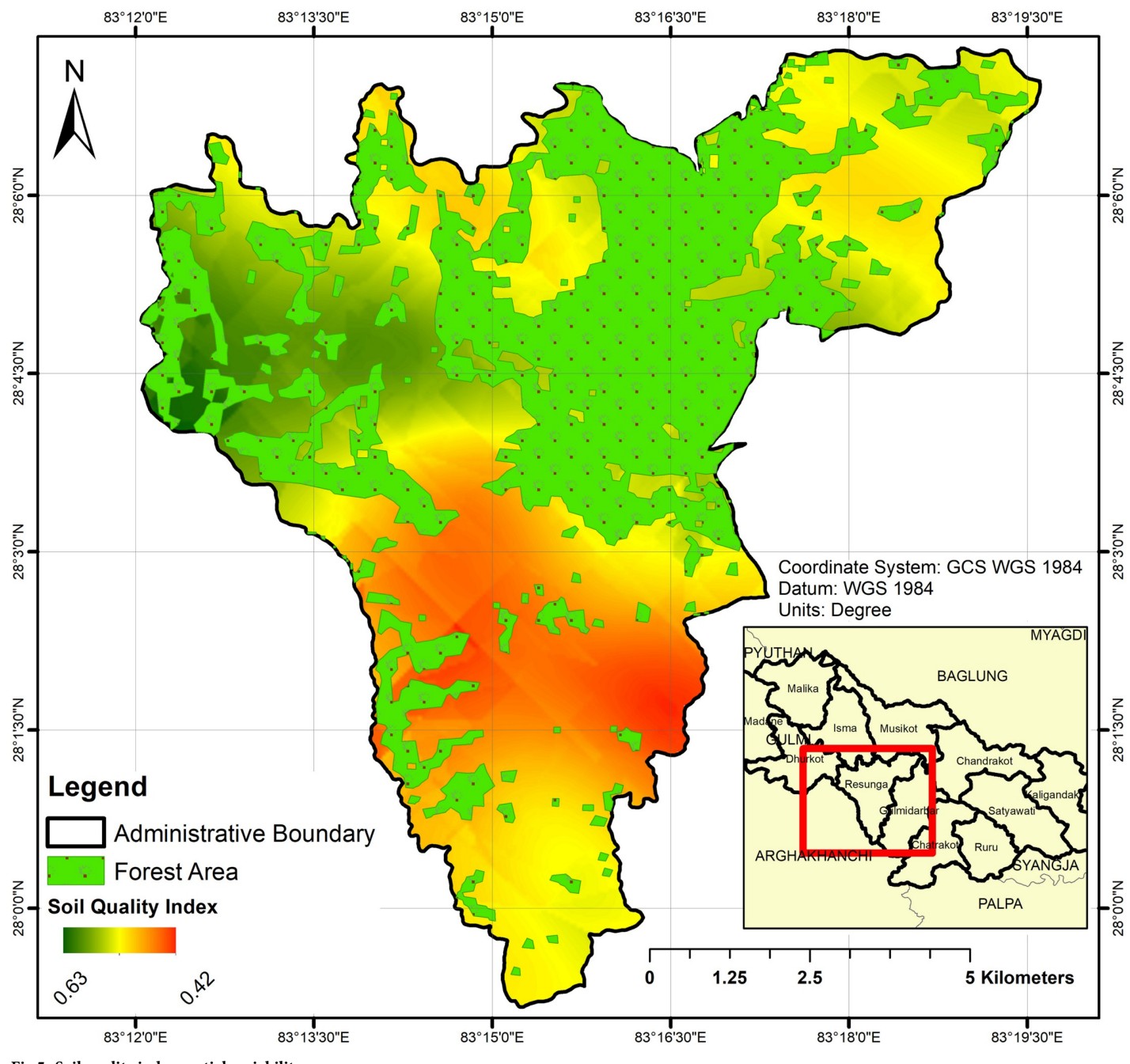

**Fig 5. Soil quality index spatial variability map.**

and produce digital maps, as well as soil quality and nutrient indices. The study revealed that areas had mostly fair and some good soil quality index SQI values. Soil organic matter and nitrogen had low levels and moderate variation in most of the area phosphorous and potassium had medium levels in most of the area. Zinc was low and boron was medium in most of the area.

Soil pH influences the soil physio-chemical, and biological properties and impacts plant growth and yield [32]. The adoption of different soil management practices could be the

**Table 6. Soil nutrient index data of Resunga Municipality, Gulmi, Nepal.**

| S. N. | Parameter | Total sample n(T) | Soil nutrient index (SNI) | Remarks |
|-------|-----------|-------------------|---------------------------|---------|
| 1 | N | 57 | 1.368 | Low |
| 2 | $P_2O_5$ | 57 | 1.842 | Medium |
| 3 | $K_2O$ | 57 | 2.053 | Medium |
| 4 | SOM | 57 | 1 | Low |
| 5 | B | 57 | 1.439 | Low |
| 6 | Zn | 57 | 1 | Low |

Note: If NI<1.67, low; NI from 1.67–2.33, medium; NI>2.33, high.

reason for a variation in the soil pH. Furthermore, the variability in the soil pH could be due to the nature of parent materials, microtopography and the types and duration of fertilizer used [33]. Soil acidity is caused by the release of $H^+$ ions during the transformation and cycling of carbon, nitrogen, sulfur, and fertilizer reactions [34]. The normal pH range for optimal mineral element availability and sustained crop production is 6.0 to 7.5 [35]. The majority of cropland in Nepal has less than 2.2% of organic matter status [36]. The low status of organic matter is due to inappropriate adaptation of sustainable soil management practices. The above-ground crop residue is used as animal feed which then affects its incorporation into the soil [37]. Incorporating the crop residue in the soil increases the fertility and organic matter percentages[38]. Insufficient balancing of chemical fertilizer application with the organic matter has been observed, whereby the recommended rate of 2.5 to 3 tons ha$^{-1}$ of organic matter application for fertility management in Nepal has not been able to keep up with the increased use of chemical fertilizers [39]. Furthermore, the mineralization rate also due to high temperature can cause low organic matter in the soil. The low status of soil organic matter is an indication of soil degradation and soil erosion [40]. The variability in total N content is due to variations in soil management practices, application of fertilizer, and Farm Yard Manure (FYM) in the study area. The deficiency in the total soil nitrogen is mainly due to low organic matter content, mineralization, and inadequate application of nitrogen to high nitrogen-demanding plants [33]. Nitrogen content is significantly low in the cultivated area compared to forest and grazing land [41]. Nitrogen is highly susceptible to loss through ammonia volatilization, runoff, and leaching [42]. Soil phosphorous is most affected by soil pH, the ideal value between 6 and 7.5 is best for phosphorous availability [43] while pH below 5.5 and between 7.5 and 8.5 decreases phosphorous availability in soil due to aluminium, iron, or calcium fixation. Phosphorous does not leach out of the root zone, erosion and runoff are the possible causes of P loss. The application of organic matter and placement of phosphorus fertilizer also impact the availability of phosphorus [44]. The availability and variability of phosphorous depend upon mean annual temperature (MAT) and mean annual precipitation (MAP), available phosphorous in soil decreases significantly with increasing MAT and MAP [45]. The medium to

**Table 7. Test of significance of pattern analysis of selected parameters.**

| Parameters | N | $P_2O_5$ | $K_2O$ | pH | SOM | B | Zn |
|------------|---|----------|--------|-----|------|----|----|
| Moran's I Index | 0.025 | -0.042 | 0.0081 | 0.112 | -0.011 | 0.050 | 0.037 |
| Variance | 0.0068 | 0.0069 | 0.007 | 0.006 | 0.0068 | 0.007 | 0.0062 |
| Z-Score | -0.088 | -0.291 | 0.310 | 1.586 | 0.007 | 0.819 | 0.703 |
| P-Value | 0.929 | 0.77 | 0.756 | 0.112 | 0.941 | 0.412 | 0.481 |

high status of potassium may be due to the low availability of calcium ions in the soil and the overall pH status of the soil was slightly acidic to neutral (5.3 to 7.5) with a variability percentage of 6.30%. The soil acidity is crucial for the availability of potassium in the soil thus reducing the calcium and magnesium ion in the soil [46]. The application of organic fertilizer and phosphorous has significantly increased the availability of potassium in soil [47]. The zinc and boron status of Nepalese soils is low in concentration and its deficiency is widespread and acute [48]. A similar type of variability in soil properties is also observed in research conducted in Zimbabwe, where variation in soil properties ranges from 11.94% to 121.99% [22]. B is available in the form of uncharged boric acid or borate to plants depending upon the soil conditions like soil moisture, temperature, pH, salinity, organic matter, and rainfall [49]. Boron deficiency is recorded as a problem and accounts for 80–90% of Nepalese soil. Zinc deficiency is widespread, and it affects 20–50% of agricultural land in Nepal [48]. The B and Zn deficiency in Nepal is due to the mismanagement of the soil rather than a general deficiency [12]. The variable affecting the micronutrient availability is pH, zinc availability rises with the decreasing pH [50, 51] and boron becomes less available with increasing pH [52]. Soil quality indices help to determine the soil quality of a particular location or ecosystem and enable the comparison between different management practices and land use [53]. The SQI was assessed so that the management issues are not only focused on crop productivity but soil degradation and environmental problems [54]. Soil quality may be affected by land use and agricultural management practices due to alterations in the physical, chemical, and biological properties of soil which result in a change in productivity [55].

The result showed that the study area lies within the fair (0.4 to 0.6) and good (0.6 to 0.8) range of the soil quality index SQI representing 96% and 3% respectively. Soil organic matter and nitrogen showed moderate variability exhibiting a low status in 95% and 86% of the total study area. Phosphorous and potassium showed medium status in 88% and 75% of the study area, respectively. Zinc was low and boron status was medium in most of the area. Based on 57 sampled data, maps of soil fertility and quality covering a 52 km$^2$ area were found to be representative in developing the spatial variability of soil attributes in non-sampled sites. The ordinary kriging method in interpolation from the geostatistical wizard is effective for determining the spatial variability of soil nutrients and is recommended for further soil mapping.

These variations have direct implications for sustainable soil management. This type of intensive soil sampling, analysis, and mapping should be conducted at the ward level at regular intervals to get a clear picture of the soil fertility status of farmland. However, in the case of highly variable soil properties, there are uncertain field-scale estimates, so a field-specific recommendation for small landholders is not feasible [56]. Soil nutrients and quality maps aid farmers, scientists, and students in decision-making based on existing soil conditions and recommend soil test-based fertilizer recommendations for intensive and sustainable crop cultivation. Soil quality and fertility maps are useful for site-specific nutrient management and soil health monitoring for SDGs and are a must for all ward and municipality levels for sustainable present and future agricultural approaches.

The study has some limitations that need to be acknowledged. The number and distribution of soil sampling locations may not be sufficient to capture the full spatial variability of soil fertility in the study area given the complexity of terrain in hilly areas. The soil properties measured in this study may not reflect all aspects of soil health, such as biological and physical properties. The ordinary kriging method used to interpolate the soil properties may introduce some errors or uncertainties in the soil fertility map. Therefore, the scope of this study is limited to providing a preliminary soil fertility map for Resunga Municipality, which can be used as a reference for soil and land use management. Future studies are needed to validate the accuracy and reliability of the soil fertility map, explore the factors influencing soil fertility in

this region, and evaluate the impact of soil and land use management practices on soil health and sustainability.

## Supporting information

**S1 File. Soil fertility mapping activities in Resunga Municipality, Gulmi, Nepal.** (**a**) Citrus farm in one of the soil sampling areas; (**b**) Farmers with a citrus plant in the study area; (**c**) Handheld GPS used in the field; (**d**) Soil sample collection and profile study; (**e**) Preparation of laboratory equipment; (**f**) Preparation of soil samples; (**g**) Laboratory analysis of soil samples. (PDF)

## Acknowledgments

The authors would like to acknowledge the Agriculture Knowledge Center, Gulmi for providing the technical support during the study and laboratory support by Soil and Fertilizer Testing Laboratory Sundarpur, Kanchanpur, Lalit BC for the technical support, Saurav Marahatta and Bala Sharma during the preparation of this manuscript, and #Mentor4Nepal Initiative for providing the support for research guidance and publication.

## Author Contributions

**Conceptualization:** Prabin Ghimire, Ashok Acharya.

**Data curation:** Prabin Ghimire, Santosh Shrestha.

**Formal analysis:** Prabin Ghimire, Santosh Shrestha, Tri Dev Acharya.

**Investigation:** Aayushma Wagle.

**Methodology:** Prabin Ghimire, Santosh Shrestha, Ashok Acharya.

**Software:** Prabin Ghimire.

**Supervision:** Tri Dev Acharya.

**Validation:** Prabin Ghimire, Santosh Shrestha.

**Visualization:** Santosh Shrestha, Tri Dev Acharya.

**Writing – original draft:** Prabin Ghimire, Aayushma Wagle.

**Writing – review & editing:** Tri Dev Acharya.

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
