## [Decision Letter · Decision Letter 0]

14 Feb 2023

PONE-D-23-01627Soil Fertility Mapping of a Cultivated Area in Resunga Municipality, Gulmi, NepalPLOS ONE

Dear Dr. Acharya,

Thank you for submitting your manuscript to PLOS ONE. After careful consideration, we feel that it has merit but does not fully meet PLOS ONE’s publication criteria as it currently stands. Therefore, we invite you to submit a revised version of the manuscript that addresses the points raised during the review process.

We look forward to receiving your revised manuscript.

Kind regards,

Surajit Mondal, PhD

Academic Editor

PLOS ONE

“This research received no external funding. The APC was funded by the University of California Davis Open Access Fund (UCD-OAF).”

5. We note that 1,3,4 and 5 in your submission contain [map/satellite] images which may be copyrighted. All PLOS content is published under the Creative Commons Attribution License (CC BY 4.0), which means that the manuscript, images, and Supporting Information files will be freely available online, and any third party is permitted to access, download, copy, distribute, and use these materials in any way, even commercially, with proper attribution. For these reasons, we cannot publish previously copyrighted maps or satellite images created using proprietary data, such as Google software (Google Maps, Street View, and Earth). For more information, see our copyright guidelines: http://journals.plos.org/plosone/s/licenses-and-copyright.

a. You may seek permission from the original copyright holder of 1,3,4 and 5 to publish the content specifically under the CC BY 4.0 license. 

Additional Editor Comments:

The article described the spatial variability of different soil properties or nutrient in a municipality of Nepal.

1. The country is situated in the Himalayan belt and elevation of places changes greatly. Authors must clarify how sampling was done for sloping land and from what position.

2. Reference needed for the classification system of different parameters.

3. The result section needs to be grouped. Currently its looking very discrete.

4. A manuscript needs to checked thoroughly for typology errors.

Please find the attached manuscript for more corrections.

Reviewers' comments:

Reviewer's Responses to Questions

**Comments to the Author**

1. Is the manuscript technically sound, and do the data support the conclusions?

Reviewer #1: Yes

Reviewer #2: Yes

2. Has the statistical analysis been performed appropriately and rigorously? 

Reviewer #1: Yes

Reviewer #2: Yes

3. Have the authors made all data underlying the findings in their manuscript fully available?

Reviewer #1: Yes

Reviewer #2: Yes

4. Is the manuscript presented in an intelligible fashion and written in standard English?

Reviewer #1: Yes

Reviewer #2: Yes

5. Review Comments to the Author

Reviewer #1: Just few minor changes are suggested and the manuscript is recommended for minor revision. The manuscript is well written but has to follow the journals guideline. It is suggested that the authors make the correction and it can be published.

Reviewer #2: Title: The title of the article is accurate and directly relates to the purpose of the research.

The article investigates Soil Fertility Mapping of a Cultivated Area in Resunga Municipality, Gulmi, Nepal. It is an interesting approach and provide useful information for the readers. However, some improvements are necessary and more details is needed to add so as considered accepted in the PLOS ONE Journal.

The abstract should contain a clear statement of recommendations in such studies. I suggest to rewrite the abstract.

Introduction: Finally, specify the main purpose of the work and highlight the main hypotheses. In its present form, there is a double repetition of the same thread.

In the last paragraph of the introduction clearly state the novelty of the current study, according to the author knowledge and the missing/gap in the literature.

In the study area chapter add some details about the climate and particularly the annual precipitation and temperature pattern.

Materials and Methods: Refine the class boundaries.

figures 3, and 5 is not readable, legend much enlarge

Discussion: The discussion is very poor in the context of research in this area by other authors and the proportion in the structure of the work. There are two possible solutions in this case: 1) broadening with the use of the relations already indicated in the Results chapter, 2) Merging the chapters Results and Discussion with the approval of plos one editor.

Conclusions: The conclusions are constructive. Please just specify the summary of the results as in the abstract (see the manuscripts).

AbdelRahman M. A. E., Rehab H. H., Yossif T. M. H. (2021) Soil fertility assessment for optimal agricultural use using remote sensing and GIS technologies. Applied Geomatics. https://doi.org/10.1007/s12518-021-00376-1

AbdelRahman, M.A.E., Natarajan, A., C.A. Srinivasamurty & Hegde R. Estimating soil fertility status in physically degraded land using GIS and remote sensing techniques in Chamarajanagar district, Karnataka, India. Egypt. J. Remote Sensing Space Sci. (2016), http://dx.doi.org/10.1016/j.ejrs.2015.12.002

AbdelRahman, M.A.E.; Metwaly, M.M.; Afifi, A.A.; D’Antonio, P.; Scopa, A. Assessment of Soil Fertility Status under Soil Degradation Rate Using Geomatics in West Nile Delta. Land 2022, 11, 1256. https://doi.org/10.3390/land11081256

Technical Notes

I suggest to create a scheme/figure that better explain the interconnections between the several sections/subsections and between the methodologies proposed and the several results.

The conclusions section must be improved with policy recommendations, practical implications and future research.

Recommends that the POLS ONE Editorial Board publish the article taking these minor changes into account.

6. PLOS authors have the option to publish the peer review history of their article (what does this mean?). If published, this will include your full peer review and any attached files.

Reviewer #1: **Yes: **Dr Pushpa Singh

Reviewer #2: **Yes: **Mohamed A. E. AbdelRahman

---

## [Author Response · Author response to Decision Letter 0]

13 Aug 2023

Dear Reviewers, 

Pleae find the attached documents as detailed response to all the comments for the revised manuscript.

Thanks, 

Tri Dev

---

## [Decision Letter · Decision Letter 1]

14 Sep 2023

Soil fertility mapping of a cultivated area in Resunga Municipality, Gulmi, Nepal

PONE-D-23-01627R1

Dear Dr. Acharya,

We’re pleased to inform you that your manuscript has been judged scientifically suitable for publication and will be formally accepted for publication once it meets all outstanding technical requirements.

Kind regards,

Surajit Mondal, PhD

Academic Editor

PLOS ONE

Additional Editor Comments (optional):

Reviewers' comments:

Reviewer's Responses to Questions

**Comments to the Author**

1. If the authors have adequately addressed your comments raised in a previous round of review and you feel that this manuscript is now acceptable for publication, you may indicate that here to bypass the “Comments to the Author” section, enter your conflict of interest statement in the “Confidential to Editor” section, and submit your "Accept" recommendation.

Reviewer #2: All comments have been addressed

2. Is the manuscript technically sound, and do the data support the conclusions?

Reviewer #2: Yes

3. Has the statistical analysis been performed appropriately and rigorously? 

Reviewer #2: Yes

4. Have the authors made all data underlying the findings in their manuscript fully available?

Reviewer #2: Yes

5. Is the manuscript presented in an intelligible fashion and written in standard English?

Reviewer #2: Yes

6. Review Comments to the Author

Reviewer #2: (No Response)

7. PLOS authors have the option to publish the peer review history of their article (what does this mean?). If published, this will include your full peer review and any attached files.

Reviewer #2: **Yes: **Mohamed A. E. AbdelRahman

---

## [Editor Report · Acceptance letter]

21 Sep 2023

PONE-D-23-01627R1 

Soil fertility mapping of a cultivated area in Resunga Municipality, Gulmi, Nepal 

Dear Dr. Acharya:

I'm pleased to inform you that your manuscript has been deemed suitable for publication in PLOS ONE. Congratulations! Your manuscript is now with our production department. 

Kind regards, 

on behalf of

Dr. Surajit Mondal 

Academic Editor

PLOS ONE